# Learning a Meaningful Representation with Prior Domain Knowledge Applied to X-ray Sensing in Geology

**Matthew Dirks**
Department of Computer Science
University of British Columbia
Vancouver, BC V6T 1Z4, Canada
mcdirks@cs.ubc.ca

## Abstract

Mining companies typically process all the material extracted from a mine site using processes which are extremely consumptive of energy and chemicals, but by sorting valuable ore from waste rock we can reduce costs and help preserve the environment. X-ray Fluorescence sensors produce spectra data that can be used to quantify element concentrations in rock samples. Our goal is to train a model to convert X-ray Fluorescence spectra into element concentrations automatically using a small dataset of rock samples that have ground truth associated with them via geochemical assay. In machine learning, when not enough data is available (often due to increasingly complex models) or the quality of the data is insufficient, then prior domain knowledge from experts can be incorporated to guide the learner. Many applications have access to prior domain knowledge, but it's not always clear how to utilize it. We introduce a system that encodes X-ray Fluorescence spectra into a meaningful low-dimensional representation via a variation of an AutoEncoder where the decoder is a forward simulator that incorporates prior domain knowledge. We train the AutoEncoder unsupervised, then swap the decoder with a module that outputs the desired prediction targets via supervised learning. We evaluate our method with and without pre-training and compare to the baseline quantitative method. We show that our method better estimated element concentrations using the pre-trained AutoEncoder with incorporated prior domain knowledge.

## 1 Introduction

Mining companies typically process all the rocks extracted from a mine site using processes which are extremely consumptive of energy and chemicals. We can improve mining operations through a better understanding of the mined material. In particular, we can sort valuable ore from waste rock automatically which will reduce costs and help preserve the environment because the amount of material sent to further downstream mining processes is reduced. Detecting the composition of rocks will lead to these improvements.

We have been working with domain experts in geology, mineralogy, and physics on the task of quantifying element concentrations in rocks from spectra measurements via X-ray Fluorescence (XRF) sensors (Muller (1972); Van Grieken & Markowicz (2001); Beckhoff et al. (2007)) mounted to mining equipment to perform real-time sensing. An XRF spectrum is a series of energies (measured in keV) and measured photon counts. Our sensor produces 1024 bands (between 0 and 33 keV). An example of an XRF spectrum from a rock is shown in Figure 1.

Fully quantitative methods typically require a set of pure reference specimen ("standards") to establish calibration from photon counts to element concentrations which is costly and time-consuming. Our goal is to perform quantitative XRF analysis at mine sites without requiring a costly calibration phase. Instead, the model is trained using rock samples taken directly from the mine, some of which are then geochemically assayed to obtain ground truth for supervised machine learning.

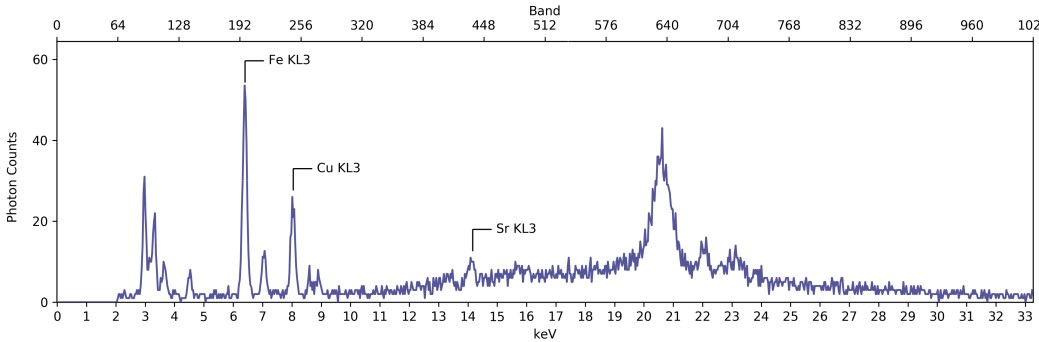

Figure 1: One example of an XRF spectrum from our dataset. This one has a significant amount of iron (Fe) and copper (Cu), and just enough strontium (Sr) to be obvious to a trained eye.

## 2 PRIOR KNOWLEDGE

Supervised learning is standard practice in AI and Machine Learning where a model is trained using a very large dataset of annotated or labelled data. While the exact number of instances needed varies, it is known that more complex models require more training instances (Beleites et al. (2013); Figueroa et al. (2012)). Roughly speaking, it is recommended to have "hundreds of thousands"[1] or more. But what if, for some task, you don't have much data? There are a few options: (1) get more data, (2) change the model to one that needs less data (usually a simpler one), and (3) teach the machine in some other way. Option 1 is not feasible when it is difficult (or costly) to get more annotated data. Option 2 is only feasible for easy problems where simple models perform well. Therefore, we want to pursue option 3. More specifically, we can better teach the machine by incorporating prior knowledge (Yu et al. (2010); Niyogi et al. (1998); Zhou & Tan (2015); Teng et al. (2015); Rudin & Wagstaff (2014)).

## 3 OUR METHOD: SPECTROSCOPY AUTOENCODING

We discussed the problem with domain experts (in geology and physics) and collected all kinds of details regarding XRF sensing. From this prior knowledge we built a simulator that reconstructs a spectrum. Many proprietary and some open-source XRF simulator software exists that simulate XRF spectra with great precision under well-defined conditions (Ritchie (2009); Brigidi & Pepponi (2017)). However, we need to adapt to noisy environments and only need to approximate XRF's most relevant features because ultimately we use machine learning to learn the remaining complexities and interactions. Furthermore, we require the ability to differentiate through the simulation end-to-end, so we built a custom simulator using TensorFlow.

AutoEncoders are commonly used to build low-dimensional representations (Bengio et al. (2013)). Our method, Spectroscopy AutoEncoding, encodes spectra from X-ray Fluorescence into a meaningful low-dimensional representation via a variation of an AutoEncoder where our XRF simulator is used in place of the usual decoder. Some prior knowledge is incorporated via architecture choices in the encoder: a 1D convolution layer captures structure in the spectrum and fully-connected layers learn the interactions between related elements. However, the majority of prior knowledge is incorporated into the simulator. Our method learns a representation which, as input into the simulator, has defined meanings. Namely, 34 variables correspond to elements amounts (measured in photon counts) and 1 variable represents the amount of background scatter in the spectrum. Parameters of the simulator (not specific to individual rocks) are also learnt during training. AutoEncoder training is unsupervised, requiring only examples of XRF spectra. After pre-training the AutoEncoder, we swap the simulator with a neural network that maps from the outputs of the pre-trained encoder to the desired prediction targets (element concentrations) via supervised learning.

---

[1]Jeff Dean, head of the Google Brain team, https://goo.gl/AWfsrK

Wu et al. (2017) use a similar concept of learning a representation using a simulator, but they recover a physical world representation from images and feed it to a physics simulator followed by a graphics engine to generate images.

# 4    EVALUATION AND RESULTS

We evaluate our method (Spectroscopy AutoEncoding) with and without pre-training and compare to the baseline quantitative method using 232 rock samples that have geochemical assays for ground truth. In testing our method without pre-training, we skip the unsupervised AutoEncoder phase and instead train directly the supervised phase which uses the encoder module but does not use our XRF simulator. This scenario is designed to check how much worse performance is when we skip learning a meaningful low-dimensional representation in pre-training. We use the standard empirical calibration method to quantify element concentration from XRF spectra as a baseline.

We found that our method, Spectroscopy AutoEncoding (with pre-training), performs better than the baseline on 7 out of 11 elements. Our method without pre-training behaved similarly to with pre-training for elements Mg, Cu, Fe, Al, and S but performed much worse than both the baseline and our method with pre-training on all remaining elements. Thus, learning the low-dimensional representation with prior knowledge was crucial to our method's performance.

The baseline performed best on 3 elements and tied with ours for element Ti. For target elements where the baseline was better than our method, ours (with pre-training) was only slightly worse. We plotted scatter plots of ground truth vs predictions and observed that our method appears to perform quite well on all target elements despite the numeric metrics reported above. The 3 elements—Sr, Ba, and P—where the baseline scored better RMSE don't look any better than our method by eye suggesting that other metrics may be needed to characterize performance. We tried mean absolute error, mean squared error, and Pearson correlation coefficient ($R^2$) but these all had similar problems.

# 5    DISCUSSION AND CONCLUSION

This paper establishes a prototype system for estimating element concentrations using individual rocks that have been crushed into powder. Ultimately, our target application is to estimate element concentrations directly inside the mine via sensors mounted on huge 120 metric ton mining shovels.[2] The environment inside open-pit mines and underground is harsh, noisy, and dirty. In this setting, rock samples cannot be prepared, distance and angle to the sensor varies, and the atmosphere (air, dust, rain, snow) will interfere. Data is also more difficult (and expensive) to collect. Thus, future work will take advantage of the unsupervised nature of AutoEncoders to pre-train a model that can then be fine-tuned on a small set of annotated data while incorporating as much prior knowledge as possible.

Regarding incorporating prior knowledge, there are some open research questions we would like to address in the future: Can we automatically refine a model given new prior knowledge? How do we best handle all the various forms of prior knowledge that experts provide? Can we use an expert's prior knowledge to improve their trust of the system?

In this paper we proposed a novel method for quantitative XRF analysis, and demonstrated a method for prior knowledge incorporation given limited annotated data. Our method shows promise to be used in automating XRF calibration and analysis in the noisy uncontrolled environments found in large-scale mining operations.

ACKNOWLEDGMENTS

Thanks to my academic supervisor, David Poole, for guidance and support; David Turner, David Munoz-Paniagua, Peter How, and others for their domain expertise; and to MineSense Technologies Ltd. for use of their sensors and rocks samples.

---

[2] For example, a P&H 4800.

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
