# OpenReview forum: "Learning a Meaningful Representation with Prior Domain Knowledge Applied to X-ray Sensing in Geology"
_ICLR.cc/2018/Workshop — Reject_

### Official Review · AnonReviewer3 · 2018-03-05
**Real world application incorporating domain knowledge into autoencoders.**

**Rating:** 6
**Confidence:** 2

**Review:**

The paper uses an autoencoder with pre-training in order to incorporate prior knowledge, applied to analysis of rocks for mining using X-ray fluorescence sensors. There is enough novelty in the application and the method, and the setting is clearly provided.

Pros:
- interesting methodology and appliation
- promising results

Cons:
- "collected all kinds of details regarding XRF sensing". This is a bit vague ... a couple of examples for the reader?
- results are presented very vaguely. "our method ... performs better" without any quantitive results. A table of results is a bare minimum here
- the results are presented for different elements, but there is no way to know from this if detection of
some of the elements is in general much easier; this makes it hard to assess the results

---

### Official Review · AnonReviewer2 · 2018-03-07
**Review to "Learning a Meaningful Representation with Prior Domain Knowledge Applied to X-ray Sensing in Geology"**

**Rating:** 3
**Confidence:** 5

**Review:**

This work presents a framework to detect the nature of the ground based on X-Ray sensing. The purpose is to prevent expensive and environmentally costly direct chemical testing. The author proposed in order to address small dataset issue to use an unsupervised training stage that performs dimension reduction on XRF spectrum to feed a later supervised stage.

Pros:
- The field of application is very interesting and Machine Learning yet has to prove its ability in it.
- The author presents the pre-training stage as dimension reduction but he actually performs an inversion of a known model (a simulator) which seems to be a good way to introduce prior knowledge by unsupervised learning

Cons:
- No information on the data split (Train / Valid / Test), how hyper parameters have been tuned ?
- No information on the AE architecture used at the pre-training stage
- No information on the ANN architecture used at the supervised stage
- Zero numerical results

The following fact does not influenced my judgment but I find surprising to see "We" all over the text nevertheless the article has only one author.

---

### Official Review · AnonReviewer1 · 2018-03-11
**Application of representation learning to mining, unclear how generalizable results are**

**Rating:** 3
**Confidence:** 3

**Review:**

The task is to estimate element concentrations from X-ray spectra of rock samples. A priori information is invoked to identify relevant representations using a physical simulator.

Pros
* Interplay of physical models and machine learned representation is a relevant topic for ICLR
* Conclusion seem to indicate that the learned representations are useful.

Cons
*  Description is vague, lacks precision (objective functions, model complexity etc etc)
*  Results are anecdotal
* Conclusion is weak

---

### Decision · Program_Chairs · 2018-03-20
**ICLR 2018 Workshop Acceptance Decision**

**Decision:**

Reject

**Comment:**

Based on the reviews, this paper has not been accepted for presentation at the ICLR workshop. However, the conversation and updates can continue to appear here on OpenReview.